# Cysteine Sulfoxides Enhance Steroid Hormone Production via Activation of the Protein Kinase A Pathway in Testis-Derived I-10 Tumor Cells

**DOI:** 10.3390/molecules25204694

**Published:** 2020-10-14

**Authors:** Yuya Nakayama, Hsin-Jung Ho, Miki Yamagishi, Hiroyuki Ikemoto, Michio Komai, Hitoshi Shirakawa

**Affiliations:** 1Health Care Research Center, Nisshin Pharma Inc., Saitama 356-8511, Japan; ikemoto.hiroyuki@nisshin.com; 2Research Center for Basic Science, Research and Development, Quality Assurance Division, Nisshin Seifun Group, Inc., Saitama 356-8511, Japan; yamagishi.miki@nisshin.com; 3Laboratory of Nutrition, Graduate School of Agricultural Science, Tohoku University, Sendai 980-8572, Japan; hsinjung@hs.hokudai.ac.jp (H.-J.H.); mkomai@m.tohoku.ac.jp (M.K.); shirakah@tohoku.ac.jp (H.S.); 4Faculty of Health Sciences, Hokkaido University, Sapporo 060-0812, Japan

**Keywords:** cysteine sulfoxides, onion, testosterone, progesterone, protein kinase A (PKA)

## Abstract

Testosterone plays an important role in male sexual characteristics and maturation, and decreased testosterone levels increase the risk of several diseases. Recently, onion extract rich in cysteine sulfoxides, which are amino acids unique to onions, has been reported to alleviate age-related symptoms resulting from decreased testosterone levels in males. However, the mechanism underlying the suppression of low testosterone levels by cysteine sulfoxides has not been elucidated. In this study, we found that onion extract containing cysteine sulfoxides enhanced progesterone, a precursor of testosterone, in mouse testis-derived I-10 tumor cells. Furthermore, cysteine sulfoxides activated protein kinase A (PKA) and cyclic adenosine monophosphate response element-binding protein, which are key factors in steroidogenesis. These results suggest that cysteine sulfoxides enhance steroid hormone production via activation of the PKA signaling pathway.

## 1. Introduction

Testosterone plays an important role in fetal development [1], sperm production [2], and the development of male sexual characteristics. However, testosterone production is known to decrease with aging and obesity [3,4,5]. Decreased testosterone levels can cause infertility, sexual dysfunction, and age-related symptoms known as late-onset hypogonadism (LOH) in older male populations [6,7]. Low testosterone blood levels can also predict the development or exacerbation of type 2 diabetes, cardiovascular disease, metabolic syndrome, and depression [8,9,10,11].

Testosterone is produced and released from Leydig cells in the testicles [12]. Progesterone is an important hormone that is converted into various steroids, including testosterone [13]. Luteinizing hormone (LH), which is secreted from the pituitary gland, stimulates testosterone production via the LH receptor (LHR) in Leydig cells. The LHR, a G-protein-coupled receptor, activates adenylate cyclase (AC) and elevates intracellular cyclic adenosine monophosphate (cAMP) levels. cAMP-dependent protein kinase A (PKA) is a key protein for testosterone production after the stimulation of LH [14]. Some food ingredients have been reported to enhance testosterone production in testis-derived tumor cells via modulation of PKA activity [15,16].

Cysteine sulfoxides (propenyl-l-cysteine sulfoxide (PCSO; isoalliin), S-methyl-L-cysteine sulfoxide (MCSO; methyiin), and cycloalliin (CA)) are unique amino acids in onions that are well known to have various health functions [17,18,19]. The oral administration of onion juice to rats has been reported to increase plasma testosterone significantly [20]. A randomized double-blind placebo-controlled parallel-group comparative clinical trial found that onion extract containing a high level of cysteine sulfoxides suppressed decreases in testosterone levels and alleviated age-related symptoms [21]. However, the mechanism and functional constituents underlying this effect have not been elucidated.

Therefore, in this study, we analyzed the effect of cysteine sulfoxide from onion extract on steroid hormone production in mouse testis-derived I-10 tumor cells.

## 2. Results

### 2.1. Onion Extract Stimulates Progesterone Production

To determine whether onion extract containing a high level of cysteine sulfoxides enhances steroid hormone production, I-10 cells were incubated with onion extract (0.3, 1, and 3 mg/mL) for 0, 3, 9, 18, and 24 h. Onion extract significantly increased progesterone, a precursor of testosterone, in cultured medium in a dose-dependent manner (Figure 1). In addition, progesterone levels were increased in a time-dependent manner, with the maximum observed after 9 h. These findings indicate that onion extract rich in cysteine sulfoxides enhances steroid hormone production in I-10 cells; 3 mg/mL of the onion extract contained 400 μM PCSO, 150 μM MCSO, and 700 μM CA, respectively. In the following experiment, we analyzed the effect of 9 h of treatment with cysteine sulfoxides on progesterone production in I-10 cells.

### 2.2. Cysteine Sulfoxides Stimulate Progesterone Production

Treatment with CA (700 μM) tended to increase progesterone levels in cultured medium in I-10 cells (*p* = 0.057), while treatment with PCSO (400 μM) and MCSO (150 μM) significantly increased progesterone levels (Figure 2). Onion extract and cysteine sulfoxides did not show cytotoxicity in tested concentrations in I-10 cells (Appendix A). Cysteine sulfoxides (PCSO, MCSO, and CA) significantly increased progesterone levels in a dose-dependent manner (Figure 3). In addition, a mix of cysteine sulfoxides (PCSO 400 μM, MCSO 150 μM, and CA 700 μM) equivalent to 3 mg/mL onion extract significantly increased progesterone levels in I-10 cells. These results indicate that the constituent cysteine sulfoxides (PCSO, MCSO, and CA) of onion extract enhance steroid hormone synthesis.

### 2.3. Cysteine Sulfoxides Regulate the Activation of PKA and cAMP Response Element-Binding Protein (CREB)

Next, we analyzed the effect of cysteine sulfoxides on the activation of PKA. Treatment with forskolin and cysteine sulfoxides (PCSO, MCSO, CA, and their mixture) for 180 min significantly enhanced PKA phosphorylation levels in I-10 cells (Figure 4). Treatment with H89, an inhibitor of PKA, abolished the enhancement of progesterone production by cysteine sulfoxides (Figure 5). PKA phosphorylates several proteins involved in intracellular signaling pathways, such as CREB. Treatment with forskolin and cysteine sulfoxides (PCSO, MCSO, CA, and their mixture) for 180 min significantly enhanced CREB phosphorylation levels in I-10 cells (Figure 6). These results suggest that cysteine sulfoxides promote steroid synthesis via activation of PKA and CREB.

## 3. Discussion

Testosterone synthesis in the testes is tightly regulated by the hypothalamic –pituitary–testicular axis in the endocrine system. In this study, we found that onion extract and its constituent cysteine sulfoxides (PCSO, MCSO, and CA) significantly increased progesterone, a precursor of testosterone, in I-10 cells (Figure 3 and Figure 4), suggesting that cysteine sulfoxides can stimulate steroidogenesis in Leydig cells. On the other hand, the ingestion of onion extract rich in cysteine sulfoxides ameliorates age-related decreases in blood testosterone levels [21]. These results suggest that cysteine sulfoxides in onion extract directly enhance steroidogenesis in the testis.

Steroid hormone synthesis has been reported to be enhanced by food ingredients via the activation of PKA [15,16]. To clarify the mechanism underlying the upregulation of progesterone synthesis by cysteine sulfoxides, we analyzed their effect on the activation of PKA. Cysteine sulfoxides significantly increased both PKA (Figure 4) and CREB phosphorylation (Figure 6), and the enhancement of progesterone was abolished by treatment with H89 (Figure 5). CREB is a substrate of PKA, and plays a crucial role in steroidogenesis for the transcriptional regulation of steroidogenic genes such as StAR and Cyp11a. Our results indicate that cysteine sulfoxides promote steroid synthesis via the activation of PKA. In order to elucidate the mechanism of testosterone production by cysteine sulfoxides, it is necessary to investigate the expression of Cyp11a and StAR via CREB and their direct activation by PKA.

In Leydig cells, PKA is activated via both cAMP-dependent and -independent manners [22,23]. We measured intracellular cAMP levels after treatment with a mixture of cysteine sulfoxides in I-10 cells, and found that cysteine sulfoxides significantly increased cAMP levels (Appendix A). Thus, the activation of PKA by cysteine sulfoxides may be caused by the elevation of cAMP levels. Further investigation is necessary to elucidate the detailed mechanism underlying the upregulation of cAMP by cysteine sulfoxides.

Treatment with onion extract has been reported to increase plasma testosterone in mice [20] and to alleviate age-related symptoms in males, accompanied by increased salivary testosterone, in a clinical trial [21]. Although the detailed mechanism remains unclear, the results of this study suggest that cysteine sulfoxides in onion extract show these phenomena via the PKA signaling pathway and enhance steroid hormone synthesis in testis Leydig cells. In addition, cysteine sulfoxides may be involved in the conversion of testosterone to the active form of dihydrotestosterone. However, further studies that also investigate the pharmacokinetics and functions in vivo of cysteine sulfoxides are needed to elucidate these mechanisms in greater detail.

In this study, we found that cysteine sulfoxides enhanced steroid hormone synthesis via the activation of the PKA/CREB pathways in the testis. Therefore, onion extracts rich in cysteine sulfoxides could be expected to be useful as dietary supplements for the prevention and amelioration of age-dependent symptoms associated with decreased testosterone production.

## 4. Materials and Methods

### 4.1. Materials

As described in a previous study [20], the onion extract was prepared by cutting, extraction, column purification, and concentration with heat treatment prior to cutting so that they contained at least 6% cysteine sulfoxides. Nisshin Pharma Inc. (Tokyo, Japan) provided the onion extract. The onion extract was formulated to contain the following cysteine sulfoxides: 2.36% PCSO, 0.75% MCSO, and 4.13% CA. The resulting onion extract was dissolved in phosphate buffered saline (PBS) (Sigma-Aldrich, St. Louis, MO, USA). PCSO (Nagara Science Co., Gifu, Japan), MCSO (LKT Laboratories, St. Paul, MN, USA), and cycloalliin hydrochloride monohydrate (Wako Pure Chemicals, Osaka, Japan) were dissolved in PBS to prepare a stock solution (10 mM). H89 and forskolin (Sigma-Aldrich) were dissolved in dimethyl sulfoxide (Sigma-Aldrich) to prepare a stock solution (10 mM).

### 4.2. High Performance Liquid Chromatography (HPLC) Analysis

Cysteine sulfoxides (PCSO, MCSO, and CA) in onion extract were analyzed by HPLC. An ultraviolet absorptiometer (measurement wavelength 210 nm) and a polymer-based hydrophilic interaction chromatography column (Asahipak NH2P-50 4E (4.6 mm × 250 mm)) at 40 °C were used for detection. A mobile phase was prepared so that A (20 mM sodium dihydrogen phosphate, 100 mM phosphate-containing buffer solution) and B (acetonitrile) were mixed at a ratio of 19:81 (*v:v*) and performed at a flow rate of 1.2 mL/min. PCSO, MCSO, and CA were quantified by an external standard method using standard reagents. The external standard method of PCSO was established using alliin (LKT Laboratories, St. Paul, MN, USA), which is a geometric isomer of PCSO, as a substitute standard. Using this method, it was confirmed that the quantitative values of PCSO as the standard reagent matched. As the amount of PCSO that can be used for routine analysis cannot be ensured, in this study, quantification was performed by an external standard method using alliin as a substitute standard.

### 4.3. Cell Culture

Mouse testis-derived I-10 tumor cells were obtained from the Health Science Research Resources Bank (Osaka, Japan) and maintained in Ham’s F-10 medium (Sigma-Aldrich) supplemented with 10% fetal bovine serum (Gibco Life Technologies, Carlsbad, CA, USA), 50 U/mL penicillin, and 50 mg/mL streptomycin (Gibco Life Technologies, Carlsbad, CA, USA) in a 5% CO_2_ humidified incubator at 37 °C. Cells were used in experiments when they reached 80–90% confluence.

### 4.4. Measurement of Progesterone Levels in Culture Medium

I-10 cells were plated at a density of 6.0 × 10^4^ cells/well in 12-well plates and incubated overnight. The culture medium was then replaced with fresh medium containing onion extract (0.3, 1.0, and 3.0 mg/mL), PCSO (100, 400, and 1000 μM), MCSO (150, 300, and 500 μM), CA (100, 700, and 1000 μM), or MIX (PCSO 400 μM, MCSO 150 μM, and CA 700 μM). The culture medium was collected and centrifuged at 1000× *g* for 5 min after incubation for 0–24 h. Progesterone concentrations in the supernatant were determined with specific enzyme immunoassay kits (Cayman Chemical, Ann Arbor, MI, USA).

### 4.5. Cell Growth Assays

I-10 cells were plated at a density of 1.2 × 10^4^ cells/well in 96-well plates. The medium was changed the following day, and onion extract (PCSO, MCSO, CA, or MIX) was added. Following incubation for 24 h, the number of viable cells in each sample was determined using the Premix WST-1 Cell Proliferation Assay System (Takara Bio Inc., Shiga, Japan) according to the manufacturer’s instructions.

### 4.6. Western Blot Analysis

I-10 cells were lysed using lysis buffer (50 mM HEPES-NaOH (pH 7.5), 150 mM NaCl, 10% glycerol, 1% Triton X-100, and 1.5 mM MgCl_2_) containing proteinase inhibitors (Complete Proteinase Inhibitor Cocktail, Roche Applied Science, Mannheim, Germany) and phosphatase inhibitors (PhosSTOP Phosphatase Inhibitor Cocktail, Roche Applied Science) to obtain whole cell extracts. The protein concentrations of the whole cell extracts were measured using a protein assay reagent (Takara Bio Inc., Shiga, Japan). The proteins were loaded onto a sodium dodecyl sulfate (SDS) gel loading buffer, resolved by 10% SDS-polyacrylamide gel electrophoresis, and transferred onto a polyvinylidene fluoride membrane (Millipore, Billerica, MA, USA). Membranes were blocked for 1 h in tris buffered saline with tween 20 (TBS-T) (10 mM Tris-HCl (pH 7.4), 150 mM NaCl, and 0.1% Tween 20) containing 5% bovine serum albumin (Sigma-Aldrich) and incubated with anti-phosphorylated PKA or anti-phosphorylated CREB (Cell Signaling Technology, Danvers, MA, USA). Membranes were visualized with the Immobilon Western reagent (Millipore) using a LAS-4000 mini luminescent image analyzer (Fujifilm, Tokyo, Japan). The relative expression levels were normalized according to the amount of α-tubulin detected by their respective antibodies (Sigma-Aldrich and Abcam, Tokyo, Japan).

### 4.7. Statistical Analysis

Data are expressed as mean ± standard deviation (SD) and analyzed using Student’s *t*-test or one-way analysis of variance followed by Dunnett’s multiple comparison test using JMP^®^11 (SAS Institute, Cary, NC, USA). *p* values < 0.05 were considered significant.

## 5. Conclusions

Onion extract rich in cysteine sulfoxides, which are unique amino acids found in onions, has been reported to alleviate age-related symptoms in males caused by decreased testosterone levels, referred to as LOH. However, the mechanism underlying the action of cysteine sulfoxides against LOH has not been elucidated. Therefore, this study investigated the underlying mechanism and functional constituents for increased steroid hormone synthesis using onion extract containing cysteine sulfoxides in mouse testis-derived I-10 tumor cells. The results indicated that cysteine sulfoxides significantly enhanced steroid synthesis via activation of PKA; cysteine sulfoxides also activated CREB. Therefore, cysteine sulfoxides may promote steroid synthesis via PKA and CREB activation in the testes.

## Figures and Tables

**Figure 1 molecules-25-04694-f001:**
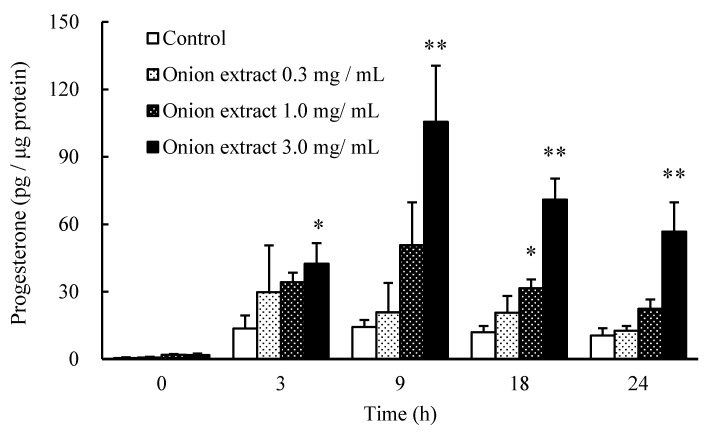
Effect of onion extract on progesterone production in I-10 cells. Data are presented as mean ± standard deviation (*n* = 3). * *p* < 0.05, ** *p* < 0.01 vs. control group.

**Figure 2 molecules-25-04694-f002:**
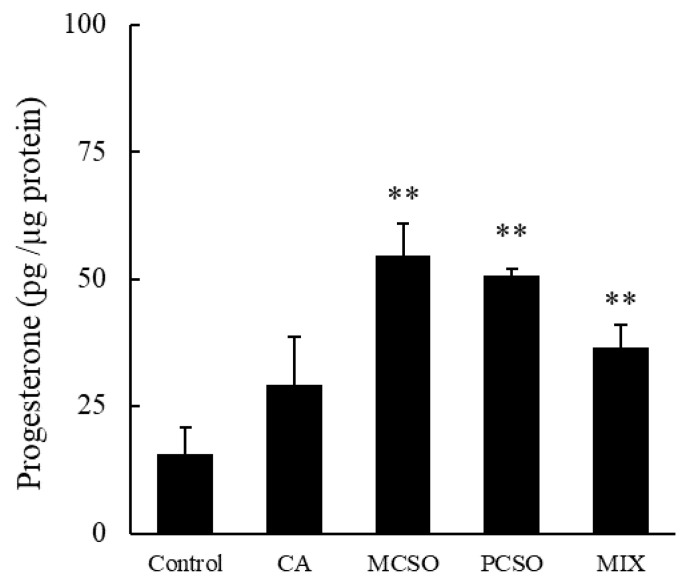
Effects of PCSO (400 μM), MCSO (150 μM), CA (700 μM), and MIX (PCSO 400 μM, MCSO 150 μM, and CA 700 μM) on progesterone production in I-10 cells. Data are presented as mean ± standard deviation (*n* = 3). ** *p* < 0.01 vs. control group. PCSO: propenyl-l-cysteine sulfoxide; MCSO: S-methyl-L-cysteine sulfoxide; CA: cycloalliin.

**Figure 3 molecules-25-04694-f003:**
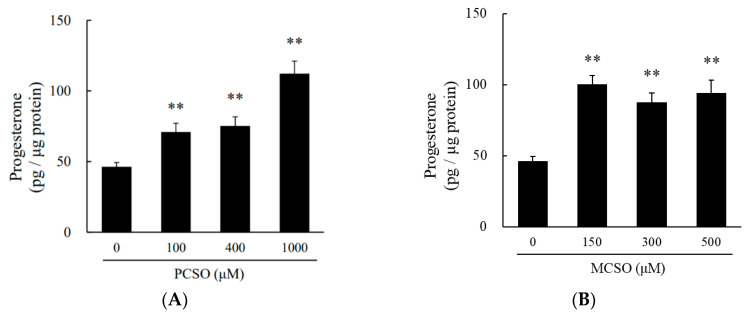
Effects of (**A**) PCSO, (**B**) MCSO, and (**C**) CA on progesterone production in I-10 cells. Data are presented as mean ± standard deviation (*n* = 3). ** *p* < 0.01 vs. control group. PCSO: propenyl-l-cysteine sulfoxide; MCSO: S-methyl-L-cysteine sulfoxide; CA: cycloalliin.

**Figure 4 molecules-25-04694-f004:**
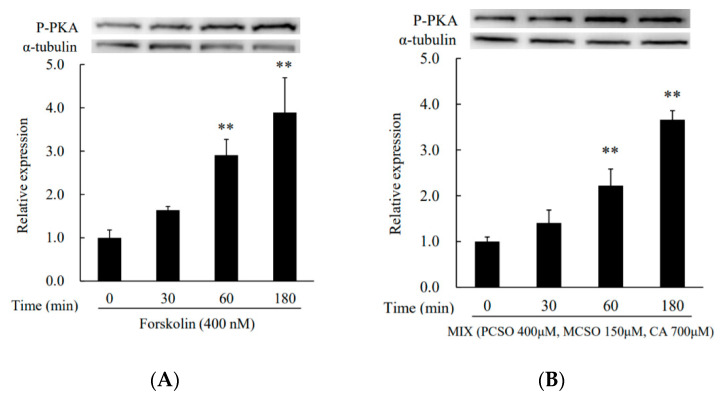
Effects of (**A**) forskolin, (**B**) MIX, (**C**) PCSO, (**D**) MCSO, and (**E**) CA on PKA activation in I-10 cells. Data are presented as mean ± standard deviation (*n* = 3). * *p* < 0.05, ** *p* < 0.01 vs. control group. PCSO: propenyl-l-cysteine sulfoxide; MCSO: S-methyl-L-cysteine sulfoxide; CA: cycloalliin; PKA: protein kinase A.

**Figure 5 molecules-25-04694-f005:**
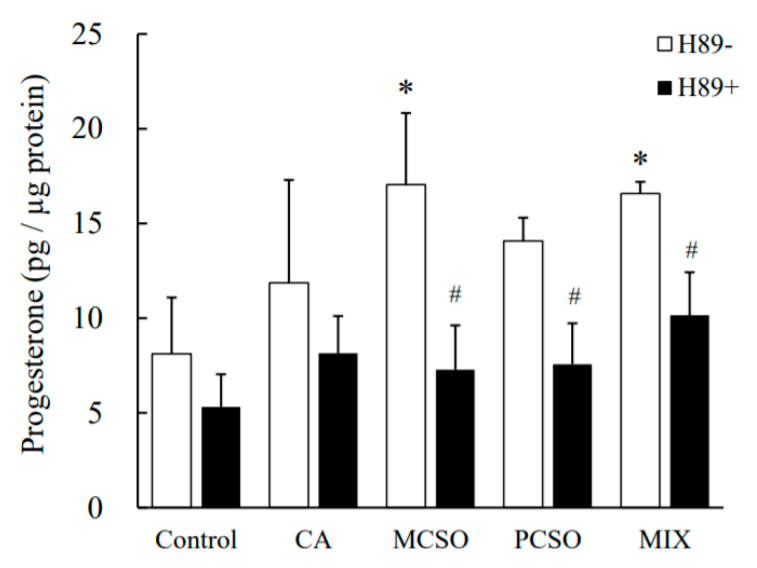
The enhancement of progesterone production by PCSO, CA, MCSO, and MIX with H89, an inhibitor of PKA, in I-10 cells. Data are presented as mean ± standard deviation (*n* = 3). * *p* < 0.05 vs. control group. ^#^
*p* <0.05 vs. H89–. PCSO: propenyl-l-cysteine sulfoxide; MCSO: S-methyl-L-cysteine sulfoxide; CA: cycloalliin.

**Figure 6 molecules-25-04694-f006:**
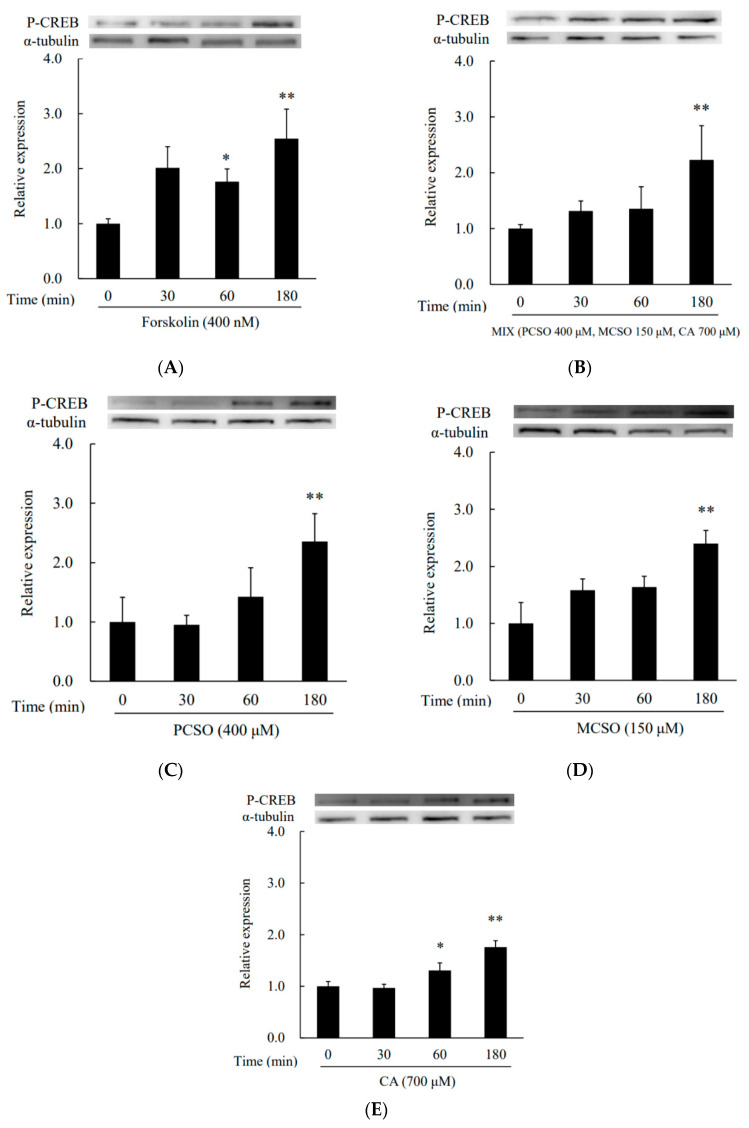
Effects of (**A**) forskolin, (**B**) MIX, (**C**) PCSO, (**D**) MCSO, and (**E**) CA on CREB activation in I-10 cells. Data are presented as mean ± standard deviation (*n* = 3). * *p* < 0.05, ** *p* < 0.01 vs. control group. PCSO: propenyl-l-cysteine sulfoxide; MCSO: S-methyl-L-cysteine sulfoxide; CA: cycloalliin; CREB: cAMP response element-binding protein; cAMP: cyclic adenosine monophosphate.

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
