# Peer review of "Cysteine Sulfoxides Enhance Steroid Hormone Production via Activation of the Protein Kinase A Pathway in Testis-Derived I-10 Tumor Cells"

_molecules, 2020, doi:10.3390/molecules25204694_

Round 1

Reviewer 1 Report

Testosterone is the main sex hormone in sperm males. It plays a very important role in many processes and sexual features. The onion extract containing a high level of cysteine sulfoxides suppressed decreasing in testosterone level. An innovative approach to the subject consisted of checking the mechanism underlying the suppression in low testosterone levels by cysteine sulfoxide. All parts of the article are very well prepared. The methodology of the experiment is well planned.
The obtained results provide important information for research on the influence of various factors on the production of sex hormones.
I recommend the article for publication.

Author Response

Thank you for inviting us to submit a revised draft of our manuscript entitled, “Cysteine sulfoxides enhance steroid hormone production via activation of the protein kinase A pathway in testis-derived I-10 tumor cells” to Molecules. We also appreciate the time and effort you and the other reviewer have dedicated to provide insightful feedback on ways to strengthen our paper. Thus, it is with great pleasure that we resubmit our article for further consideration.

We believe that the manuscript is now acceptable for publication in Molecules.

Thank you for your consideration of this manuscript.

Sincerely,

Yuya Nakayama

Reviewer 2 Report

The manuscript proposed by Nakayama et al. explores the mechanism by which cysteine sulfoxide contained in onion extracts controls testosterone production in I-10 tumor cells.

- Although the study on I-10 tumor cells is well performed and the data presented are clear, some mechanistic studies should be performed (ChIP). 

- Expression level of steroidogenic genes should be performed to further decrypt the mechanism by which cysteine sulfoxide stimulates testosterone production.

- in vivo experiments on harvested testes in mouse or rat model are missing to confirm that the suggested mechanism also happens in vivo.

- Testosterone is converted to dihydrotestosterone (DHT) by the action of 5α-reductase in target tissues; although it is about one-tenth as abundant as testosterone, it accounts for most of testosterone’s biological action. It would be of interest to see if the increase testosterone production converts to dihydrotestosterone.

- L130: “decreases”

- L140: some words are missing in the sentence.

Author Response

Point 1: Although the study on I-10 tumor cells is well performed and the data presented are clear, some mechanistic studies should be performed (ChIP) 

Response 1: Thank you for your suggestion. We also consider that various studies are necessary to elucidate the mechanism of action. We would like to proceed with the examination as a future issue. Therefore, we added “In order to elucidate the mechanism of testosterone production by cysteine sulfoxides, it is necessary to investigate the expression of Cyp11a and StAR via CREB and their direct activation by PKA.” (Line 140-142, Page 9, Revised manuscript).

Point 2: Expression level of steroidogenic genes should be performed to further decrypt the mechanism by which cysteine sulfoxide stimulates testosterone production.

Response 2: Thank you for your suggestion. We agree that some steroidogenic genes should be investigated for further elucidation of the mechanism by which cysteine sulfoxide stimulates testosterone production. Therefore, we added “In order to elucidate the mechanism of testosterone production by cysteine sulfoxides, it is necessary to investigate the expression of Cyp11a and StAR via CREB and their direct activation by PKA.” (Line 140-142, Page 9, Revised manuscript).

Point 3: in vivo experiments on harvested testes in mouse or rat model are missing to confirm that the suggested mechanism also happens in vivo.

Response 3: Thank you for your suggestion. Since we have not confirmed the transfer of cysteine sulfoxides to the testis, it is unclear whether similar results will be obtained in vivo. Therefore, we revised “Therefore, cysteine sulfoxides in onion extract can directly enhance steroidogenesis in the testis.” into “These results suggest that cysteine sulfoxides in onion extract directly enhance steroidogenesis in the testis.” (Line 130-131, Page 8, Revised manuscript) and “this” into “these” (Line 156, Page 9, Revised manuscript). And we added “and functions in vivo” (Line 155-156, Page 9, Revised manuscript).

Point 4: Testosterone is converted to dihydrotestosterone (DHT) by the action of 5α-reductase in target tissues; although it is about one-tenth as abundant as testosterone, it accounts for most of testosterone’s biological action. It would be of interest to see if the increase testosterone production converts to dihydrotestosterone.

Response 4: Thank you for your suggestion. We would also be interested in the effect on DHT. We have now acknowledged this and added it as a topic for further research. We added “In addition, cysteine sulfoxides may be involved in the conversion of testosterone to the active form of dihydrotestosterone.” (Line 153-155, Page 9, Revised manuscript).

Point 5: L130: “decreases”

Response 5: Thank you for your comment. We revised “decreases” into “decrease” (Line 130, Page 8, Revised manuscript).

Point 6: L140: some words are missing in the sentence.

Response 6: Thank you for your comment. We revised “In Leydig cells, PKA activates via both cAMP-dependent and -independent manners [22,23].” into “In Leydig cells, PKA is activated via cAMP-dependent or independent manners [22,23].” (Line 143, Page 9, Revised manuscript).

Thank you for your consideration of this manuscript.

Sincerely,

Yuya Nakayama

Round 2

Reviewer 2 Report

Although the results in the manuscript by Nakayama et al. are clearly presented and the study is well-designed, any significant additional knowledge in the field is brought by current version. None of the additional experiment suggested in the first revision has been performed.